# Destination Social Responsibility and Residents' Environmentally Responsible Behavior: Assessing the Mediating Role of Community Attachment and Involvement

Elsie Nasr [1] , Okechukwu Lawrence Emeagwali [2], Hasan Yousef Aljuhmani [3,*] and Souha Al-Geitany [4]

1   Faculty of Business and Economics, Girne American University, North Cyprus via Mersin 10, Kyrenia 99320, Turkey
2   Business Management Department, Girne American University, North Cyprus via Mersin 10, Kyrenia 99320, Turkey
3   Faculty of Business and Economics, Centre for Management Research, Girne American University, North Cyprus via Mersin 10, Kyrenia 99428, Turkey
4   Department of Business, Girne American University, North Cyprus via Mersin 10, Kyrenia 99320, Turkey
*   Correspondence: jahmani_hassan@yahoo.com

**Abstract:** This study revisited the relationship between destination social responsibility (DSR) and residents' environmentally responsible behavior (ERB) in conjunction with the stimulus–organism–response (S-O-R) framework to assess the mediating effect of community attachment and involvement. The proposed conceptual research model was empirically examined with 375 residents from the largest tourist destinations in Ghana. A cross-sectional research design was used, and structural equation modeling (SEM) was applied to test the mediating role of community attachment and involvement. The findings of this study confirmed that DSR has a significant and positive relationship with residents' community attachment, involvement, and ERB. In addition, the findings of this study revealed a positive relationship between community attachment and residents' ERB. The findings of this study also confirmed the indirect effect of DSR on residents' ERB through community attachment. Contrary to expectations, the results of this study did not support the direct and indirect effects of community involvement on residents' ERB. This study responded to the call from previous research to investigate the relationship between residents' DSR and socio-psychological constructs, such as community attachment and involvement, which in turn enhance and improve their ERB in different cultures and tourism destinations.

**Keywords:** destination social responsibility; environmentally responsible behavior; community attachment; community involvement; resident; Ghana

## 1. Introduction

Tourism destination research has emphasized the impact of corporate social responsibility (CSR) in the pursuit of creating a more sustainable tourism environment [1–5]. In order to achieve this, tourism destinations have become more reliant on the development of cultural and environmental resources, which must be developed and managed to create a more sustainable tourism environment [6]. Sustainable tourism managers need to create a more sustainable tourism environment by utilizing socially responsible practices at a tourism destination [7]. Hence, previous studies focused on the important role of destination social responsibility (DSR) in order to create a more sustainable tourism environment [2–4,6–9]. Thus, the importance of developing a more sustainable tourism environment has prompted scholarly interest in exploring the consequences of the DSR in tourism destination research [10–12].

Marketers and tourism practitioners were looking for better ways of understanding the impact of DSR on tourism research, which can be defined as "*perceptions of obligations and activities that are applied to all stakeholders, including tourists, community residents, employees, investors, governments, suppliers, and competitors*" [13]. The importance of studying the DSR concept in the tourism literature has been well identified in recent studies due to its undeniable positive consequences [14]. For instance, previous studies confirmed that DSR is positively related to tourists' environmentally responsible behavior, identification, tourism impact, satisfaction [7,11,15–17], revisit intention [2,9,11], and destination trust and image [4,18].

Accordingly, the DSR concept did not lose its importance in the tourism destination literature, and researchers have used this concept in local community studies. Thus, previous studies revealed that DSR is positively associated with different environmental factors, such as environmentally responsible behaviour [8], overall community satisfaction [8,13,19,20], community identification [13,19], quality of life [6,21–24], place attachment [12], support for tourism development [6,19,20,22,25], community commitment [20], trust and economic development [13,25], and destination sustainability [23]. Limited studies found that community-related factors, such as community attachment and involvement, could be related to environmentally responsible behavior [26–28]. Despite the growing interest in studying the possible outcomes of DSR, there is a scarcity and limited empirical studies on community DSR [8,9,29], especially from the destination residents' perspective [19]. Little is known about the influence of DSR on community-related factors, such as involvement and attachment, which in turn leads to environmentally responsible behavior (ERB). Although previous studies attempted to understand the relationship between DSR and ERB from different groups of stakeholders, such as visitors and tourists, little attention has been paid to local residents' contributions [12]. Therefore, it is necessary to investigate whether the results associated with DSR can be sustained, as the literature on this subject has been inconsistent.

In order to fill the gaps in the tourism destination literature and respond to the recent call by previous research to validate the impact of residents' DSR on other sociopsychological constructs [6], the current research was designed to investigate the relationship between DSR and community-related outcomes that have not been discovered by previous research [14]. Thus, it could be noticed that previous research did not investigate the impact of DSR on community attachment and involvement. The current study captured a considerable contribution by first validating the relationship between DSR and socio-psychological constructs (community attachment and involvement) in a tourism context. Second, by examining the impact of DSR on community attachment and involvement, the authors examined the relationship between these three factors and one crucial outcome (i.e., environmentally responsible behavior). Finally, this study was designed to draw a more comprehensive model to assess the indirect relationship between DSR and residents' ERB by understanding the mediating role of sociopsychological constructs, such as community attachment and involvement. Moreover, as proactive measures to control the outbreak of COVID-19, destination management organizations (DMOs) are required to consider what might influence domestic residents' behavior [9], which positioned this research in a timely manner. Therefore, this study is the first of its kind to explore DSR from the resident perspective in examining whether community attachment, involvement, and ERB act as outcomes of DSR.

## 2. Theoretical Framework

### 2.1. Destination Social Responsibility

The main stakeholder groups of interest in tourism destination research include but are not limited to business owners and employees, tourists, visitors, and residents [30,31]. Local residents have been identified as the main stakeholder group in several tourism destination studies [6,8,32,33]. As pointed out by Su et al. [8], the interaction of local residents with their favorable tourism destinations significantly influences their tourism

environment. As such, residents' attitudes and behaviors can be significantly affected by their interactions with the destination community [6,33–35].

Creating a more sustainable tourism environment for local residents is important for tourism destination research, and scholarly attention to the corporate social responsibility (CSR) concept remains robust [1–5]. Accordingly, Su et al. [6] pointed out that the concept of CSR that has been studied "*in the field of organizational behavior is not completely suitable to the destination context*" (p. 1041). A review of the definitions and dimensionality of residents' DSR in the tourism literature is provided in Table 1. In the current study, the main stakeholder group of interest in studying DSR was local residents' perceptions of their specific tourism destinations. Local residents' communities are considered one of the most important stakeholder groups in tourism destination research, as they can be directly affected by the losses and benefits from the tourism development level in their local communities [36–45] compared to other stakeholder groups [33,35]. Thus, "*understanding tourism development from the local resident standpoint will deepen our understanding of both the long-term success and sustainability of tourist destinations*" [46].

**Table 1.** Definitions and dimensionality of residents DSR in the tourism literature.

| | Author(s) | Definition | Dimensions |
|---|---|---|---|
| [6] | Su, Huang, and Huang (2018) | "the collective ideology and efforts of destination stakeholders to conduct socially responsible activities as perceived by local residents" (p. 1041). | Environmental Economic Social Stakeholder |
| [13] | Su et al. (2017) | "DSR is about perceptions of obligations and activities that are applied to all stakeholders, including tourists, community residents, employees, investors, governments, suppliers, and competitors" (p. 490). | Economic Environmental Social Stakeholder Voluntariness |
| [25] | Ma et al. (2013) | "the 'status and activities' applied to all its stakeholders (including tourists, employees, community residents, investors, governments, suppliers and competitors) upon the perception of its social obligations" (p. 5948). | Economic Environmental Social Stakeholders Voluntariness |

Note(s): Destination social responsibility (DSR).

Few studies have empirically examined the consequences of DSR, especially from the perspective of destination residents. Understanding the impact of DSR from the local resident's standpoint is crucial, and more research is still underway [13]. As such, this study aimed to uncover the relationship between DSR and residents' ERB by understanding the role played by community attachment and involvement.

### 2.2. Environmentally Responsible Behaviour

ERB is conceptualized as a reflection of people's concerns toward the environment, willingness to take pro-environmental action, and perceived ecological knowledge [47]. For instance, ERB is usually defined as the actions taken by individuals to solve environmental issues and protect the environment [48]. Based on the need for sustainable tourism development and environmental concerns, previous studies examined residents' ERB [8,26,49,50]. Thus, Su et al. [51] defined residents' ERB as "*behaviours taken by residents who devote themselves to minimizing adverse environmental effects and environmental protection while not destroying the environment at a destination during their day-to-day lives*" (p. 472). Generally, residents' ERB may be reflected in different behaviors such as energy management, waste recycling, composting, and sustainable transport [52–54].

Since the 1970s, tourism scholars have used a variety of methods to measure ERB [47] in different contexts, and the general agreement is that ERB is a multidimensional construct. For instance, Smith-Sebasto and D'Costa [55] suggested a multidimensional structure for

ERB and recommended that the six factors of ERB are persuasive action, physical action, legal action, financial action, educational action, and civic action. Thapa [56] classified ERB into political actions, recycling, education, green consumption, and community activism. Other studies refer to ERB as eco-friendly, low-impact, environmentally friendly, pro-environmental, green, and conservation behaviors [7,57]. At the same time, previous studies measured and conceptualized ERB from a more holistic approach and suggested a more universal measure of ERB. For example, Lee et al. [58] proposed a measure for investigating ERB from the viewpoint of community-based tourists in seven dimensions: environmentally friendly behavior, pro-environmental behavior, sustainable behavior, persuasive action, physical action, financial action, civil action, and educational action. Safshekan et al. [26] adopted this scale when studying the ERB of residents on Northern Cyprus Island by using the above seven dimensions. For this reason, this study also used Lee et al.'s [58] conceptualization to measure residents' ERB.

ERB is also known as green behavior, environmentally friendly, eco-friendly, and pro-environmental behavior [57,59]. Regardless of the different wording of ERB, these concepts are well established and used inherently in the sustainable tourism literature, emphasizing individuals' positive behaviors in the pursuit of creating a more sustainable tourism environment by protecting the natural environment [60,61]. ERB refers to residents' actions to avoid or reduce the impacts of destroying the environmental resources in destinations [51]. Various guiding frameworks, such as the stimulus–organism–response (S-O-R) theory [62], the norm activation model (NAM) [63], the theory of interpersonal behavior (TIB) [64], place attachment theory [65], the theory of planned behavior (TPB) [66], and the value–belief–norm theory (VBN) [67] are the most used theories in assessing ERB and pro-environmental behavior [68–79]. The S-O-R, TPB, and TIB are the most important and used theories to understand and assess the antecedents of residents' ERB [61,77–79]. While the vitality of other theories, such as NAM or VBN, in predicting ERB is widely supported in previous research [78,80–83].

To date, a number of antecedents of residents' ERB have been identified [49], including, but limited to, place attachment [75]; community attachment, involvement, and environmental attitudes [26]; residents' community participation [84]; psychological ownership [85]; residents' evaluation of environmental reputation and quality [51]; sustainable tourism development attitude [86]; destination social responsibility; tourism impacts; overall community satisfaction [8]; social capital [87]; and locus of control, altruism [88]. The studies mentioned above are significant in helping scholars understand how to better clarify, explain, and predict residents' ERB. However, to the best of our knowledge, no empirical study has been conducted to investigate the causal relationships between DSR, community attachment, community involvement, and residents' ERB.

### 2.3. Theory of Planned Behavior and the Theory of Interpersonal Behavior

TPB and TIB suggested that attitudes toward a behavior are directly associated with the intention to perform the behavior. The TIB framework offers a series of behavioral variables similar to those in the TPB; similar to TIB, TPB is considered a more used theory of individual behaviors able to capture different aspects of environmental and behavioral intentions [61,73,76–78]. According to Su et al. [3], previous studies employed the TPB focus on social factors and individual traits as predictors of ERB, with destination factors (e.g., DSR) being ignored. We attempted to fill this void in the literature with the aid of the TIB model [68,69]. Thus, TPB and TIB share similar value measures, such as behavioral beliefs, normative beliefs, and behavioral intentions. For example, TIB suggests that behavioral beliefs have a direct relationship with behavioral intentions, whereas TPB proposes that behavioral beliefs shape individuals' attitude toward the behavior first and then results in individual behavioral intentions [89]. A possible justification for integrating these two theories is predicated on the fact that environmental activities (e.g., DSR) have acted as the predictor of individuals' attitudes (e.g., community attachment), which ultimately leads to them acting in more responsible behavior toward their specific destination. Based

on the above discussion, proposing a positive relationship between residents' DSR and community attachment and involvement, this study examined the effect of community attachment and involvement on residents' ERB.

*2.4. Stimulus–Organism–Response Framework*

However, TPB and TIB theories alone are insufficient to investigate the predictors of residents' ERB in the sustainable tourism literature. This paper took the TPB and TIB theories in conjunction with the S-O-R framework to better understand and assess the antecedents of residents' ERB. The S-O-R model was proposed by Mehrabian and Russell [62] and then modified by Jacoby [90]. The model advocates that different environmental factors (stimulus) guide people's cognitive and emotional behavior (organism), which in turn leads to specific behavioral reactions (responses) [62]. S-O-R theory is well suited to serve as the theoretical foundation of this study for several reasons. Firstly, previous studies in the context of ERB frequently draw upon the S-O-R framework in understanding the predictors of different environmental behaviors constructs, including but not limited to ERB [3,7,76,79,91–93], pro-environmental behavior [94–96], and green behavior [49,97,98]. Secondly, the S-O-R framework provides great flexibility that enables one to assess different types of stimuli, organisms, and responses [90]. Thus, the stimulus refers to the social and physical environment that people perceive; the organism is characterized as unobservable, internal processes; and the response can be understood as individuals' attitudes and behaviors [49]. Finally, this is the most widely used framework in the sustainable tourism context [14]. Recent studies [3,7,8,15,17,18,91,99,100] adopted the S-O-R model to evaluate the impact of different external environmental factors (S) (e.g., DSR activities) on the internal states (O) (e.g., community attachment and involvement), which in turn leads to individual behavioral responses (R) (e.g., ERB).

Accordingly, the first component of the S-O-R framework, stimulus, refers to the different external environmental factors (e.g., social psychological stimuli and external object stimuli) that stimulate an individual's internal state [79,90,101]. According to Su et al. [8], DSR can be viewed as a stimulus to residents; Su et al. [6] defined it as "the collective ideology and efforts of destination stakeholders to conduct socially responsible activities as perceived by local residents." DSR is considered an important environmental factor [6,20]. In fact, residents' behaviors have been considered central and fundamental in stimulating residents' DSR initiatives, regarded as the main driver in our theoretical framework [6,8,12,19,23,102]. Moreover, an organism in the S-O-R model refers to the individual's cognitive processes and internal emotional states [79,101]. In this study, community attachment and involvement are considered as the organism state. According to Lee [103], community attachment and involvement can be regarded as an emotional bond between an individual and a community. Thus, community attachment reflects an "individual's rootedness and sense of belonging to a community" that is captured as the organism state in our study [103]. Finally, ERB is regarded as the responses of residents that may be expressed by their behavior. Considering the above, it is evident that adopting the S-O-R model is suitable.

Hence, based on the S-O-R model, the following sections aim to address a gap in the literature by developing the conceptual research model depicted in Figure 1 to uncover the relationship between DSR and residents' ERB by understanding the role played by community attachment and involvement.

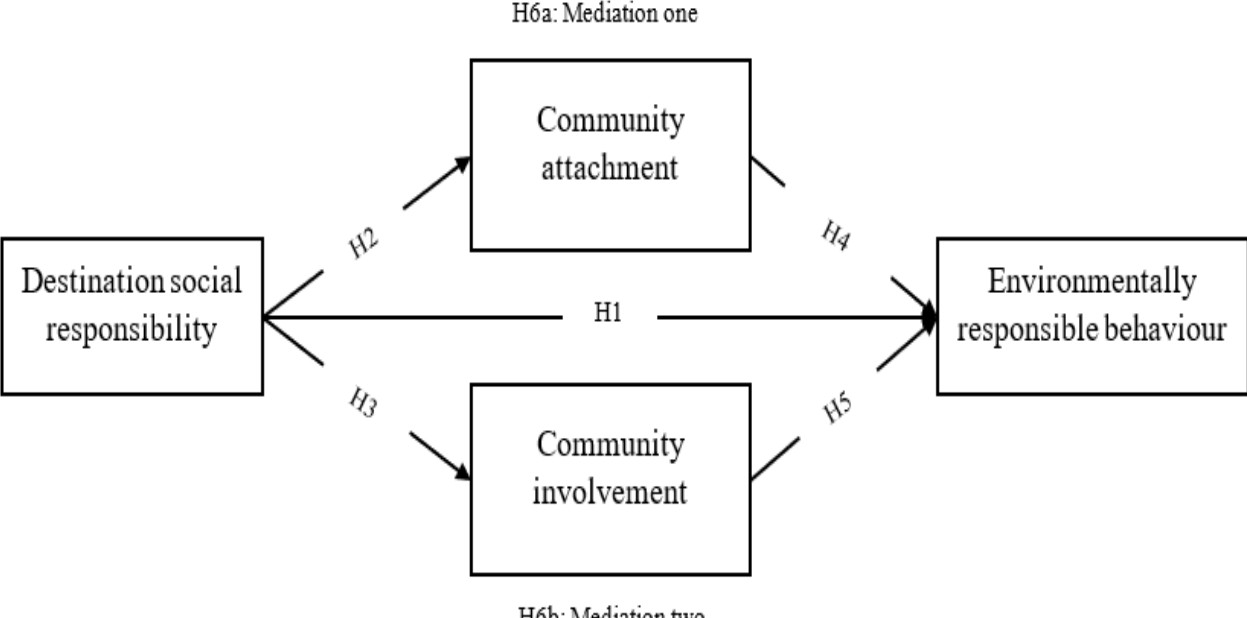

**Figure 1.** Conceptual research model. Note: H6 represents the mediation effect of community attachment and involvement through which DSR influence residents' ERB.

### 3. Hypotheses Development

*3.1. Distention Social Responsibility and Environmentally Responsible Behaviour*

Consequently, the main focus of this research was on how DSR is related to residents' ERB. Previous studies extensively focused on exploring the predictors and antecedents of residents' ERB [8,26,49,51]. These studies described ERB as a daily practice performed by local residents to maintain and protect the environment and/or reduce the negative impact on their natural environment. However, these studies suffer from inconsistencies between the conceptualization and operationalization of the ERB construct. For example, Su et al. [8] conceptualized residents' ERB as a unidimensional construct and found that DSR enhanced and improved resident ERB. Other studies suggested a multidimensional construct for ERB [47,48,53,56,58,104,105]. For example, Cottrell and Graefe [47] suggested that ERB includes ecological knowledge, commitment, and environmental concerns.

There is little consensus on the true composition of the ERB concept and how it should be measured. However, to our knowledge, there is no empirical study that provides a clear measurement of residents' ERB dimensions, and no research has examined the DSR-resident ERB relationship from a more holistic measure [8], suggesting that "*future studies should seek more reliable measures to assess residents' actual ERB*" (p. 187). As such, we investigated the posited conceptual relationships between DSR and the higher-order construct of a resident's ERB, including sustainable behavior, pro-environmental behavior, and environmentally friendly behavior conceptualized by Lee et al. [58], to examine whether DSR is a factor that must be present for residents' ERB to occur. In this regard, previous studies asserted that DSR shapes residents' behavior, which minimizes their negative impacts and generates environmental, social, and economic benefits toward their local community and environment [6,8]. Thus, DSR behaviors can improve residents' pro-environmental behavior and destination environment [6,29,30]. Therefore, we argued that DSR leads residents to show a high level of ERB. Thus, we proposed the following hypothesis:

**Hypothesis 1 (H1).** *Distention of social responsibility is positively related to residents' environmentally responsible behavior.*

### 3.2. Distention Social Responsibility, Community Attachment, and Involvement

In tourism studies, DSR can be defined as the daily activities and obligations of stakeholders toward a tourism destination [4,9,106]. These stakeholders include local residents as the primary group of stakeholders [6]. Su and Huang [106] further mentioned that stakeholders should protect and improve destination interests (e.g., economic, social, cultural, and environmental interests). Gursoy et al. [20] suggested that DSR can be measured by three subscales (environmental, social, and economic), as perceived by local residents. DSR can be classified into local economic, social, cultural, and environmental responsibilities [23]. Other studies [7,8,13,19] described DSR through a multidimensional structure comprising five dimensions (environmental, social, economic, stakeholder, and voluntary responsibilities). Researchers also explored the relationship between community attachment, defined as an "*individual's social participation and integration into community life, and reflects an affective bond or emotional link between an individual and a specific community*" [103,107]. Others looked to the community as a "*social structure*" comprising normative, institutional, and ecological dimensions [108]. Community attachment is the level of a person's sense of belonging and rootedness to a place and community [109,110].

According to Lee [103], community involvement is conceptualized as the degree to which members of the community are involved in daily and routine activities that are embedded in the communities where they live. Thus, community involvement can be defined as "*the extent to which residents are involved in sharing issues about their lives with their communities*" [103]. The way residents are involved in their communities includes activities such as participating in the process of tourism planning, self-management, marketing, employment, and decision-making [111–113]. Thus, resident community involvement has repeatedly been reported in tourism studies, and the relationship between DSR and community involvement from the resident destination perspective has not been explicitly investigated in previous studies [114]. However, tourism scholars examined the relationship between DSR and community involvement in different settings. For example, Su et al. [13] argued that in implementing DSR programs, "*tourism designers and planners should be cautious and ensure that local residents are involved.*"

However, research has not been conducted extensively, either empirically or theoretically, to examine the antecedents and outcomes of community attachment and involvement. "*What are the community attachment variables that most directly impact on residents' attitudes toward tourism development?*" [115]. This is the main question of whether DSR influences residents' community attachment and involvement, which in turn influences environmentally responsible behavior. Clearly, DSR is considered an important factor that has a significant impact on community and place and has repeatedly been used in social responsibility models by tourism scholars to examine its effects on residents' perceptions of tourism attitudes and impacts toward a particular tourism destination [6,8,9,12]. However, the results are contradictory; for example, previous research found that residents who are highly socially responsible for their environments are strongly attached to their community. Thus, we proposed the following hypotheses:

**Hypothesis 2 (H2).** *Distention social responsibility is positively related to community attachment;*

**Hypothesis 3 (H3).** *Distention social responsibility is positively related to community involvement.*

### 3.3. Community Attachment, Involvement, and Environmentally Responsible Behaviour

In tourism studies, scholars argued that community attachment could shape residents' behaviors toward tourism growth and development [116–118]. Others argued that there is a significant relationship between community and place attachment and pro-environmental behavior [75]. In particular, previous studies revealed that natural community attachment is positively related to pro-environmental behavior, whereas civic community attachment is negatively related to pro-environmental behavior [54]. However, Safshekan et al. [26]

and Nugroho and Numata [119] asserted that community attachment has an insignificant influence on residents' environmental attitudes and support for tourism development.

Regarding the role of community involvement in pro-environmental behavior, previous studies found that positive emotional attachment and identity-based attachment positively affect residents' actual actions and behavioral intentions [120]. Attitude–behavior theory states that individual behaviors are the direct consequences of their attitudes [26,121,122]. Previous research discovered that community and place are antecedents of environmentally responsible behavior [123–130]. Prior studies mainly focused on the association between familiar residential places and people's environmental behaviors [131,132]. For example, they also stated that people who are highly attached to their community and place have a commitment to their local environment, which in turn improves and enhances their environmentally responsible behavior on a daily basis [131]. Other studies argued that when people are attached to their tourist locations, they are responsible for environmental issues and are concerned about environmental protection [125,133,134]. Recent studies also revealed that involvement is positively and significantly related to tourists' environmentally responsible behavior [26,52,105,135]. Others also pointed out that individuals with a high level of involvement in environmental activities were more likely to select eco-friendly service providers than individuals with a lower level of involvement [52,136–138].

In particular, previous studies indicated inconsistent results regarding the relationship between community and place and pro-environmental behavior [75]. For example, some studies provided either positive [69,131], negative [139,140], or null [117,141] relationships between community attachment and environmentally responsible behavior. As the research results have been mixed, the following hypothesis is proposed.

**Hypothesis 4 (H4).** *Residents' community attachment is positively related to environmentally responsible behavior;*

**Hypothesis 5 (H5).** *Residents' community involvement is positively related to environmentally responsible behavior.*

*3.4. The Mediation Effect of Community Attachment and Involvement*

When considering the earlier discussion, several studies investigated the relationship between the DSR and ERB [7,8,12,91,142]. Su et al. [8] found that DSR enhances residents' ERB. They also asserted that the relationship between DSR and residents' ERB was mediated by overall community satisfaction [8]. In addition, the relationship between community attachment and residents' environmental behaviors and attitudes has also been examined in recent studies [26,27,76,127,143–145]. Vaske and Kobrin [125] and Orgaz-Agüera et al. [146] found that community and place attachment can directly or indirectly affect residents' support for tourism development and ERB. Regarding the impact of community involvement on ERB, previous studies demonstrated that residents and tourists who are highly involved with their hosted communities contribute to more environmentally responsible behavior in daily practices [26,52,136,138,147,148]. However, "*few studies have investigated the intervening mechanisms by which residents lend their support to tourism development*" [149].

As results on the relationship between DSR and ERB remain scarce, and there is a dearth of empirical validation, this study used residents' community attachment and involvement as the underlying mechanisms through which DSR influences residents' ERB (Figure 1). More recently, Lee et al. [29] examined the relationship between DSR and the pro-environmental behavior of visitors and found that the effect of high DSR on pro-environmental behavior is stronger than that of low DSR. In addition, Lee et al. [29] investigated the link between personal norms and pro-environmental behavior using DSR as a moderator. They found that the effect of personal norms on pro-environmental behavior decreased when environmental DSR increased. Hassan and Soliman [9] argued that during the COVID-19 pandemic, DSR was positively associated with destination reputation. Su and Huang [106] found that destination reputation mediates the relationship between

DSR and destination identification. Hu et al. [12] assessed the mediating effect of place attachment on the relationship between DSR and residents' ERB. They confirmed that the positive relationship between DSR and ERB was significantly mediated by place attachment. Thus, prior studies claimed that place and community attachment play a mediating role in environmentally responsible behavior [50,131,143]. Although the research results have been mixed, the following hypotheses coincide with common thinking about the mediating effect of both residents' community attachment and involvement through which DSR influences ERB. Thus, we proposed the following hypothesis:

**Hypothesis 6a (H6a).** *Residents' community attachment positively mediates the relationship between DSR and ERB;*

**Hypothesis 6b (H6b).** *Residents' community involvement positively mediates the relationship between DSR and ERB.*

## 4. Methodology

### 4.1. Data Collection and Sample

In order to test and validate the aforementioned research hypotheses, quantitative and cross-sectional research approaches were used to collect data using a self-administered questionnaire. As mentioned before, the main objective of this research was to examine the relationship between DSR, community attachment, and involvement, which in turn leads to residents' ERB. The data of this research were collected from local residents of Accra, Kumasi, and Sekondi-Takoradi, tourist destinations in Ghana (Figure 2). These tourist destination cities are famous for their unique natural scenery, rural tourism, and heritage tourism and attract a large number of tourists every year [150–153]. These cities have also been engaged in the development of rural tourism, protection of ecological culture, and poverty alleviation, as well as in providing jobs for local residents to increase their wealth and income [154–157]. Convenience sampling was used to collect data from the local residents of Ghana. A total of 800 self-administered questionnaires were distributed, and 428 were returned. After eliminating incomplete surveys, 375 usable questionnaires were collected and used for further analysis, resulting in a response rate of 46.9%.

Table 2 shows that the respondents comprised more females (53.9%) than males (46.1%). Of the respondents, 54.1% were within the age group of 21–30 years, followed by 31–40 years (28.3%) and over 40 years (14.9%). The majority of the respondents had college and university education (70.4%), with 13.9% of the sample having obtained a postgraduate degree. Then, the monthly household income of GHS 1000–1999 GHS (Ghanaian cedi) was dominant (42.9%), followed by less than GHS 1000 (29.9%), and 27.1% were more than GHS 2000 thousand. Single respondents (52.5%) were predominant, as compared with married (36.5%). The majority of respondents were from Accra (57.1%), followed by Kumasi (23.2%) and Sekondi-Takoradi (18.9%). The occupation of the respondents varied and was reasonably distributed across all occupational levels.

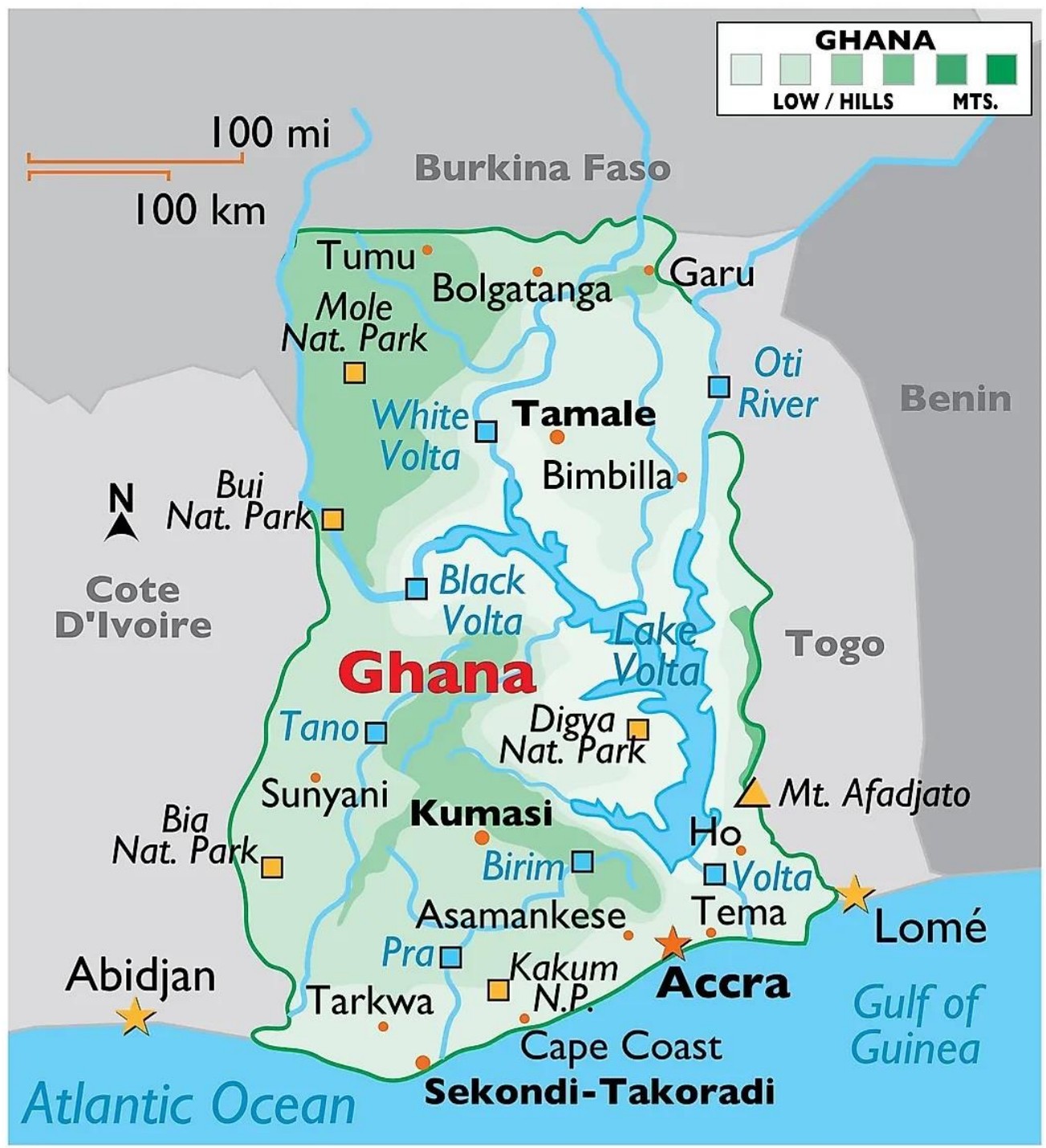

**Figure 2.** Map of Ghana (showing Accra, Kumasi, and Sekondi-Takoradi locations). Source: Worldatlas [158].

**Table 2.** Demographics of the research sample.

| Measures | Item | Frequency | Percentage (%) |
|---|---|---|---|
| Gender | Male | 173 | 46.1% |
| | Female | 202 | 53.9% |
| Age | Below 20 years old | 10 | 2.7% |
| | 21–30 | 203 | 54.1% |
| | 31–40 | 106 | 28.3% |

**Table 2.** *Cont.*

| Measures | Item | Frequency | Percentage (%) |
|---|---|---|---|
| | 41–50 | 36 | 9.6% |
| | 51–60 | 17 | 4.5% |
| | Above 60 years old | 3 | 0.8% |
| Education | No formal education | 6 | 1.6% |
| | High school | 45 | 12.0% |
| | Diploma | 78 | 20.8% |
| | University degree | 186 | 49.6% |
| | Postgraduate | 52 | 13.9% |
| | Other | 8 | 2.1% |
| Marital status | Single | 197 | 52.5% |
| | Married | 137 | 36.5% |
| | Divorce | 28 | 7.5% |
| | Widowed | 12 | 3.2% |
| | Others | 1 | 0.3% |
| Monthly income | Less than 999 GHS | 112 | 29.9% |
| | 1000–1999 GHS | 161 | 42.9% |
| | 2000–2999 GHS | 65 | 17.3% |
| | 3000–3999 GHS | 17 | 4.5% |
| | 4000–4999 GHS | 12 | 3.2% |
| | 5000 GHS or more | 8 | 2.1% |
| Occupation | Civil servant or teacher | 170 | 45.3% |
| | Student | 95 | 25.3% |
| | Office worker | 66 | 17.6% |
| | Housewife | 19 | 5.1% |
| | Service worker | 7 | 1.9% |
| | Retired | 3 | 0.8% |
| | Other | 15 | 4.0% |
| Number of family | 1 person (self) | 73 | 19.5% |
| | 2 people | 200 | 53.3% |
| | 3 people | 91 | 24.3% |
| | 4 people | 9 | 2.4% |
| | More than 5 people | 2 | 0.5% |
| Residency | Under 3 years | 31 | 8.3% |
| | 3–6 years | 57 | 15.2% |
| | 6–9 years | 19 | 5.1% |
| | 9–12 years | 39 | 10.4% |
| | 12–15 years | 55 | 14.7% |
| | 15–18 years | 34 | 9.1% |
| | 18–21 years | 47 | 12.5% |
| | 21–24 years | 48 | 12.8% |
| | 25 years and above | 45 | 12.0% |
| Ethnicity | Accra | 214 | 57.1% |
| | Kumasi | 87 | 23.2% |
| | Sekondi-Takoradi | 71 | 18.9% |
| | Others | 3 | 0.8% |
| Total | | 375 | 100% |

*4.2. Power Analysis Check*

Before analyzing the collected data to test the study hypothesis, a power analysis was calculated using G*Power 3.1.9.7 program [159] to check whether the collected sample size of this study was sufficient to represent the study population [160]. A priori analysis test showed that a minimum of 127 sample size could be considered representative and sufficient to achieve a statistical power of 0.80 for the structural model at a significant level of 0.05 with a medium effect size of 0.15 [161]. Hence, the collected sample size of 375 respondents used to test the study hypotheses is greater than the required sample size.

### 4.3. Measures

The scales for the study constructs were obtained from the related literature. DSR was measured using six items adapted from previous studies [6–9]. A sample item included "Ghana seems to include environmental concerns in its operations." Community attachment was assessed using 5 items adapted from previous research [103,119,144,149,162,163]. A sample item is "I am very attached to this community". Community involvement was measured using 4 items also adopted from previous studies [103,119,138,162,163]. A sample item included, "I am involved in the decision-making for the sustainable tourism of this community". Twelve items were used to measure the three dimensions of ERB adopted from previous research [26,58]. Specifically, sustainable behavior was operationalized through five measures. A sample item included "I observe the history and culture heritage detailed". Pro-environmental behavior was assessed using three items. A sample item included "I voluntarily stop visiting a favorite spot if it needed to recover from environmental damage". Environmentally friendly behavior was measured using four items. A sample item included "After a picnic, I leave the place as clean as it was originally". All measures used a five-point Likert scale ranging from (1 = strongly disagree) to (5 = strongly agree).

In order to ensure the accuracy of the study constructs and formulation of items, a pilot study with 30 residents of Accra, Kumasi, and Sekondi-Takoradi cities of Ghana was conducted. Participants in the pilot study indicated that there were no issues with the questionnaire or the readability of the study items. Finally, we asked two tourism faculty members to check the content validity and review the measurement items, and we asked them whether these items measured the intended constructs. Therefore, the self-administered questionnaire was not modified.

## 5. Data Analysis

### 5.1. Measurement Model and Model Fit Measures

This study tested the proposed research model using AMOS 24.0; the data analysis procedure was as follows. First, confirmatory factor analysis was conducted to verify the measurement model. Second, this study tested the hypotheses using structural equation modeling (SEM) to confirm the causal relationship between the study constructs [164]. As such, we use covariance-based SEM because it is increasingly embraced in marketing management research in general and hospitality and tourism research in particular [165,166]. The fit indices of the measurement model (Table 3) suggest that the model fits the data well ($X^2$/df = 2.30 < 3; CFI = 0.94 > 0.90; SRMR = 0.06 < 0.08; RMSEA = 0.05 < 0.08). Following Hu and Bentler's [167] evaluation criteria of the structural model, all reported fit measures were acceptable.

**Table 3.** Model fit measures.

| Measures | Recommended Criteria | Measurement Model | Structural Model | References |
|---|---|---|---|---|
| CMIN | - | 539.371 | 23.683 | |
| DF | - | 234 | 18 | |
| /d.f$^2$X | <3 | 2.30 | 1.32 | |
| CFI | >0.9 | 0.94 | 0.98 | |
| NFI | >0.8 | 0.90 | 0.97 | Hu and Bentler's [167]. |
| IFI | >0.8 | 0.93 | 0.99 | |
| TLI | >0.8 | 0.92 | 0.97 | |
| SRMR | <0.08 | 0.06 | 0.02 | |
| RMSEA | <0.08 | 0.05 | 0.03 | |

### 5.2. Reliability and Validity Tests

In order to test the constructs' reliability and validity, Cronbach's alpha and composite reliability measures were used. The results in Table 4 confirm that Cronbach's alpha values of the study constructs were all above the suggested cutoff point of 0.70 [168] and ranged

from 0.814 to 0.899. In addition, the composite reliability (CR) measures of the study constructs were all higher than the recommended threshold value of 0.60 [169] and ranged from 0.825 to 0.901. Therefore, these results demonstrate a satisfactory internal reliability measure of the study items that were used to measure the study constructs [170].

**Table 4.** Scale's measurement, reliability, and validity.

| Construct/Indicators | Factor Loadings | Cronbach's Alpha Values | CR | AVE |
|---|---|---|---|---|
| Destination Social Responsibility (DSR) | | 0.899 | 0.901 | 0.607 |
| DSR1: "Ghana seems to include environmental concerns in its operations." | 0.791 | | | |
| DSR2: "Ghana seems to give back to the local community." | 0.598 | | | |
| DSR3: "Ghana seems to be successful in their profitability." | 0.855 | | | |
| DSR4: "Ghana seems to treat its stakeholders well." | 0.818 | | | |
| DSR5: "Ghana seems to be based on ethical values and beyond legal obligations." | 0.844 | | | |
| DSR6: "Ghana seems to consider health and safety issues in its operations." | 0.737 | | | |
| Community Attachment (CA) | | 0.823 | 0.834 | 0.509 |
| CA1: "I identify the living in this community." | 0.532 | | | |
| CA2: "I feel that this community is a part of me." | 0.642 | | | |
| CA3: "Living in this community says a lot about who I am." | 0.845 | | | |
| CA4: "I am very attached to this community." | 0.839 | | | |
| CA5: "I feel a strong sense of belonging to this community." | 0.658 | | | |
| Community Involvement (CI) | | 0.824 | 0.825 | 0.542 |
| CI1: "I participate in sustainable and eco-friendly tourism-related activities." | 0.740 | | | |
| CI2: "I support research for the sustainability of this community." | 0.776 | | | |
| CI3: "I am involved in the planning and management of sustainable tourism in this community." | 0.755 | | | |
| CI4: "I am involved in the decision-making for the sustainable tourism of this community." | 0.669 | | | |
| Sustainable Behavior (SUB) | | 0.814 | 0.836 | 0.565 |
| SUB1: "I understand residents' life-styles." | - | | | |
| SUB2: "I observe the history and culture heritage detailed." | 0.565 | | | |
| SUB3: "I observe the nature and wildlife detailed." | 0.833 | | | |
| SUB4: "I pick up (encourage others) litter left by other people." | 0.807 | | | |
| SUB5: "I buy (or use) local products and services in this tour." | 0.771 | | | |
| Pro-environmental Behavior (PEB) | | 0.818 | 0.837 | 0.637 |
| PEB1: "I voluntarily visit a favorite spot less if it needed to recover from environmental damage." | 0.861 | | | |
| PEB2: "I voluntarily stop visiting a favorite spot if it needed to recover from environmental damage." | 0.899 | | | |
| PEB3: "I choose products or services with eco-labels first in this tour." | 0.601 | | | |
| Environmentally Friendly Behavior (EFB) | | 0.876 | 0.886 | 0.663 |
| EFB1: "I do not intend to disturb any creature and vegetation." | 0.620 | | | |
| EFB2: "I tell my companions not to feed the animals." | 0.828 | | | |
| EFB3: "After a picnic, I leave the place as clean as it was originally." | 0.897 | | | |
| EFB4: "I don't overturn rock and dried wood arbitrarily." | 0.883 | | | |

Note: Composite reliability (CR), Average variance extracted (AVE).

As also shown in Table 4, the convergent validity measure was adequate as the factor loading of the confirmatory factor analysis (CFA) for all study items was higher than the threshold of 0.50 and statistically significant at the 0.001 level [164]. In addition, the average variance extracted (AVE) values of all the study constructs were greater than the cutoff point of 0.50 [171] and ranged from 0.509 to 0.663. These results indicate that the convergent validity measures of the study constructs and the measurement items were satisfactory.

The discriminant validity results are presented in Table 5. Following Fornell and Larcker's [171] recommendation, if the AVEs square root values of the study constructs are higher than the intercorrelations with the other values, the discriminant validity measure is adequate. As shown in Table 5, the discriminant validity measure of this study was adequate, as the AVEs square root values of the study constructs were all higher than the intercorrelations with other values.

**Table 5.** Discriminant validity of measures.

| Factors | 1 | 2 | 3 | 4 | 5 | 6 |
|---|---|---|---|---|---|---|
| 1. Destination social responsibility | **0.779** | | | | | |
| 2. Community attachment | 0.405 *** | **0.714** | | | | |
| 3. Community involvement | 0.598 *** | 0.463 *** | **0.736** | | | |
| 4. Pro-environmental behavior | 0.288 *** | 0.261 *** | 0.232 *** | **0.798** | | |
| 5. Sustainable behavior | 0.558 *** | 0.304 *** | 0.314 *** | 0.419 *** | **0.752** | |
| 6. Environmentally friendly behavior | 0.469 *** | 0.414 *** | 0.369 *** | 0.376 *** | 0.585 *** | **0.814** |

Note: square root of average variance extracted (AVE) is shown on the diagonal (in bolds) of the matrix; inter-construct correlations are shown off the diagonal; *** significant at level of 0.001.

### 5.3. Common Method Variance

After validating discriminant and convergent measures, the next step was to test the threat of having a common method variance (CMV), as all the study constructs were obtained from the same participants using the same instrument [172]. In this regard, Harman's [173] single-factor test was used as the most used indicative of CMV [174]. Therefore, all the study constructs were factored to load into an unrotated exploratory factor analysis as a single factor. Harman's test results indicated that the total variance explained for the first factor was 31.85%, which is less than the recommended cut-off point of 50%, while the eigenvalue of the single factor was greater than 1.0. As this factor did not account for the majority of the covariance between the measures, we assumed that common method bias is not a pervasive issue in this study [175]. An extraction with eigenvalues above 1.0 with varimax rotation confirmed this interpretation, as all items loaded highly on their respective scales. Therefore, we can conclude that CMV should not be a serious concern in this research.

### 5.4. Structural Path Model and Model Fit Measures

Before testing the hypotheses of the study through covariance-based SEM, the psychometric properties of constructs in the hypothesized model were also evaluated by checking for non-normal data distribution analysis using the normality estimation procedure set out in Amos [176]. In this structural model, our results show that the skewness values of the study constructs ranged from −0.82 to +1.89. Kurtosis values also ranged from −0.15 to +3.84, which is considered to be normally distributed. According to Collier's [177] recommendations, the acceptable "*skew values range between −2 and +2*" and "*kurtosis, the range is −10 to +10 to still be considered normally distributed*" (p. 166). Thus, the issue of having non-normal data distributed should not be a serious concern in this study.

The structural model testing results (Table 3) showed fit values of X2/df = 1.32, CFI = 0.98, SRMR = 0.02, RMSEA = 0.03 coefficients that were all higher than commonly accepted standards, and the model fit the data well [167].

### 5.5. Testing of Research Hypotheses

The results of the structural model analysis are presented in Table 6. The results of this study confirmed that DSR was positively and significantly related to residents' ERB (β = 0.671, *p* < 0.001), providing support for H1. Additionally, the path coefficient from DSR to community attachment (β = 0.450, *p* < 0.001) was significant and positive, indicating that H2 was supported. As expected, the relationship between DSR and community involvement was positive and significant (β = 0.666, *p* < 0.001); therefore, H3 is supported. The results revealed that the impact of community attachment on resident ERB was positive and significant (β = 0.252, *p* < 0.001), providing support for H4. Finally, community involvement had a negative and insignificant effect on resident ERB (β = −0.026, *p* = 0.549); hence, H5 was not supported. The results of the structural model tests are shown in Figure 3.

**Table 6.** Direct effect results. Note(s): destination social responsibility (DSR), environmentally responsible behaviour (ERB); *** statistically significant at $p < 0.001$.

| Direct Effect | Standardized Coefficients | Standard Errors | t-Values | p-Values | Decision |
|---|---|---|---|---|---|
| H1: DSR → ERB | 0.671 *** | 0.018 | 15.942 | 0.001 | Accepted |
| H2: DSR → community attachment | 0.450 *** | 0.028 | 9.736 | 0.001 | Accepted |
| H3: DSR → community involvement | 0.666 *** | 0.028 | 17.285 | 0.001 | Accepted |
| H4: Community attachment → ERB | 0.252 *** | 0.026 | 6.909 | 0.001 | Accepted |
| H5: Community involvement → ERB | −0.026 | 0.025 | −0.599 | 0.549 | Rejected |

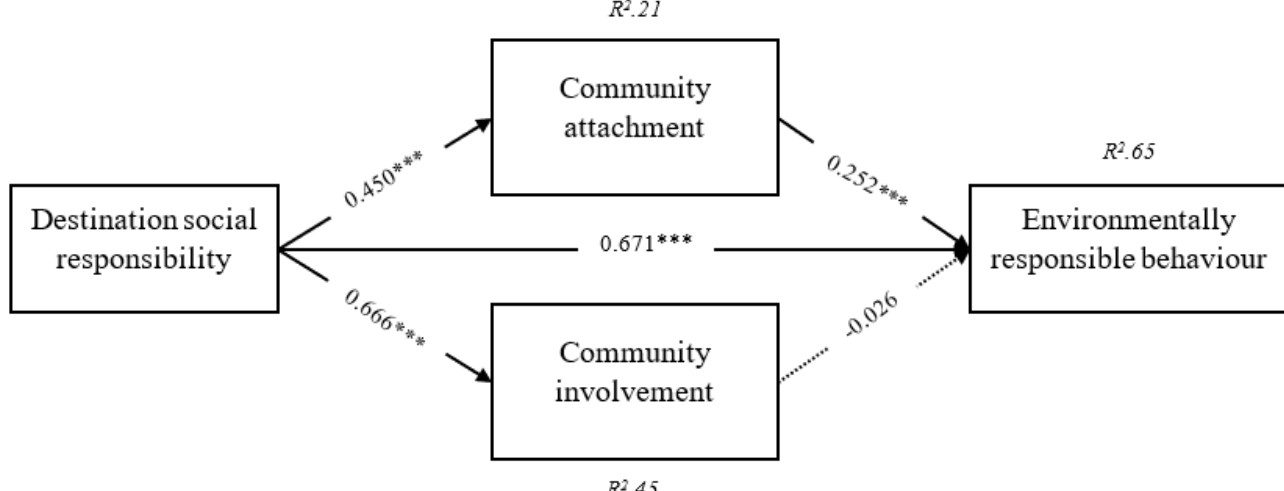

**Figure 3.** Empirical results of the conceptual model. Note: solid arrows represent statistically significant predictive relationships. Dotted arrows represent insignificant predictive relationships. *** Statistically significant at $p < 0.001$; $n = 375$.

### 5.6. The power of the Integrated Research Model

DSR explained 45% of the variance in community involvement and 21% of the variance in community attachment, whereas DSR, community involvement, and attachment explained 65% of the variance in resident ERB (Figure 3). $R^2$ values achieved an acceptable level of explanatory power, as recommended by Cohen [178], indicating a substantial model.

### 5.7. Testing the Mediating Effect of Community Attachment and Involvement

In order to assess the mediating role of community attachment and involvement through which DSR influences resident ERB, we used Baron and Kenny's [179] four-step approach and bootstrapping procedure with a recommended resample of 2000 with a 95 percent confidence interval [180]. The results of H6a support the hypothesis that community attachment mediates the relationship between DSR and ERB. However, the results in Table 7 confirm that destination social responsibility had a significant indirect effect ($\beta = 0.113$ ***) on environmentally responsible behavior through community attachment. This indicates that community attachment partially mediates the effect of DSR on resident ERB.

**Table 7.** Indirect effect result.

| Hypothesized Path | Indirect Effect | Lower Bound | Upper Bound | p-Values | Results |
|---|---|---|---|---|---|
| H6a: DSR → CA → resident ERB | 0.113 *** | 0.35 | 0.65 | 0.001 | Accepted |
| H6b: DSR → CI → resident ERB | −0.017 | −0.32 | 0.14 | 0.581 | Rejected |

Note(s): destination social responsibility (DSR), community attachment (CA), community involvement (CI), environmentally responsible behaviour (ERB); *** statistically significant at $p < 0.001$.

In contrast, we also argued that community involvement mediates the relationship between DSR and resident ERB. The results in Table 7 indicate that the indirect relationship between DSR and ERB is statistically insignificant through community involvement, and no mediating effect is demonstrated (β = −0.017, n.s.), lending no support to H6b.

## 6. Discussions and Implications

### 6.1. General Discussion

The main focus of this research is to develop a comprehensive research model based on the S-O-R framework to investigate the impacts of destination social responsibility, community involvement, and attachment on environmentally responsible behaviors at Accra, Kumasi, and Sekondi-Takoradi, tourist destinations in Ghana. Although the relationship between DSR and ERB was supported by previous studies [7,8,12,91,142], they failed to integrate the sociopsychological constructs (community attachment and involvement) through which environmental factors such as DSR influence residents' ERB [6]. Thus, this study filled the research gaps in previous studies by assessing the mediation effect of community attachment and involvement to support the linkages of DSR with residents' environmentally responsible behaviors developed by previous studies [7,8,12,19,29,142,181].

Accordingly, the concept of DSR was considered an important environmental factor that enables local residents to reduce the negative impact on the environment and to provide social and economic benefits for local residents [6,7]. Hence, tourist destinations rely greatly on different environmental factors, such as residents' DSR behaviors, which in turn enhance and improve their environmentally responsible behaviors [8]. Thus, the findings of this study confirm that DSR has a significant and positive relationship with residents' ERB (Hypothesis 1). This result is consistent with the results of previous studies [7,8,12,91,142], although in different contexts. However, to the best of our knowledge, no empirical study has been conducted to investigate the causal relationships between DSR and residents' ERB through community attachment and involvement.

Previous studies empirically investigated the impact of community attachment and/or involvement on ERB [26,27,52,76,127,143–145,148] but not the destination social responsibility with community attachment and involvement [6,114]. Furthermore, within the context of residents' DSR, no empirical research investigated the impact of DSR on community attachment and involvement [14]. However, we considered the current conceptual research model as a contribution to the existing tourism destination literature by empirically testing the causal relationship between resident DSR, community attachment, and involvement. The findings of this study show that residents' DSR has a positive impact on community attachment and involvement (Hypotheses 2 and 3). Thus, residents who are highly responsible for their tourism destinations are more attached to and involved in their local community. The way residents are involved in their communities includes activities such as participating in the process of tourism planning, self-management, marketing, employment, and decision-making [111–113]. Community attachment is reflected by an individual's rootedness and sense of belonging to the community [109,110].

The way local residents feel a sense of belonging towards a particular tourism destination depends greatly on how much they are socially responsible towards their tourism destination, which in turn enhances and improves their environmentally responsible behaviors. Hence, the findings of this research revealed a positive relationship between community attachment and residents' ERB (Hypothesis 4). This result is in line with previous studies [26,69,75,127,131,144,162,182–185], which confirmed that community and place attachment are positively related to ERB. In addition, the findings of this study contradict the results of previous studies [117,119,139–141]. Therefore, the positive and significant relationship between community attachment and resident ERB was noteworthy. Thus, community attachment could act as an intervening mechanism through which DSR influences residents' ERB.

Likewise, the findings of this study also confirmed the indirect effect of DSR on residents' ERB through community attachment (Hypothesis 6a). Therefore, residents' destination social responsibility increases community attachment, which in turn enhances and improves their environmentally responsible behavior. Thus, the mediating effect of community attachment on the relationship between DSR and ERB is complementary. This result is consistent with the arguments of previous studies [125,146,186]; improving DSR practices increases residents' ERB directly and indirectly by enhancing their sense of belonging (community attachment) toward a specific tourism destination.

Contrary to expectations, the results of this study did not support the direct (Hypothesis 5) and indirect (Hypothesis 6b) effects of community involvement through which DSR influences residents' ERB. However, the insignificant and negative relationship between community involvement and residents' ERB is inconsistent with the results of Safshekan et al. [26], Nugroho and Numata [119], and Zhu et al. [138], who found that community involvement was positively and significantly related to residents' ERB. However, we still have justification for this insignificant result. This result means that the local community of Ghana did not believe that being more socially responsible for their tourism destination was more involved in their local community, but that it did not affect their ERB activities significantly.

Accordingly, the insignificant relationship between community involvement and residents' ERB does not neglect the importance of involving local residents in ERB activities. As such, the findings of this study advance prior research by showing a partial mediation effect of DSR on residents' ERB through community attachment, which explains 65% of the total variance in ERB. Hence, residents' ERB levels were high regardless of their level of involvement. In addition, residents' ERB levels may be increased by their attachment to their local communities rather than their involvement level. The lack of ability of local residents to actively participate in their communities has been reported in previous studies [144,146,162]. "*It is also important to induce community residents' active participation in community activities*" [115]. As "*the participation of the local community offers new opportunities to residents, to mobilize their capabilities as social actors, rather than as passive subjects, so that they can make decisions and take control over the activities that affect their lives*" [146]. Therefore, local authorities could involve local residents in ERB activities to ensure long-term sustainability and to protect and improve places that are most favorable to them. They also have to reconsider the processes of engaging local communities to enhance and improve residents' community attachment and involvement in different environmental activities such as ERB.

*6.2. Theoretical Implications*

This study contributed to the literature on sustainable tourism development and tourism destinations in several ways. First, previous studies have not investigated the proposed conceptual research model. Thus, prior studies [7,8,12,91,142] assessed the relationship between destination social responsibility and environmentally responsible behavior but paid scant attention to the intervening machines through which environmental factors such as DSR influence residents' ERB [14,29,149]. This study responded to the call of previous research to investigate the relationship between residents' DSR and socio-psychological constructs such as community attachment and involvement, which in turn enhance and improve their environmentally responsible behavior in different cultures and tourism destinations. Therefore, to test and validate the proposed research model, data were collected from a developing country (Ghana) because of its unique natural scenery, rural tourism, and heritage tourism [150,152,153]. Data were collected from local residents to validate the mediation effect of community attachment and involvement in the relationship between DSR and ERB. Thus, this research provides valuable and important findings in the existing literature.

Second, the current study also contributed to the responsible tourism and sustainable tourism literature by empirically investigating the impact of DSR on community attachment and the involvement of local residents. The results of this study confirmed that residents' DSR increased the community attachment and involvement of local residents.

Thus, previous studies supported the impact of residents' DSR on other environmental factors, such as resident identification, emotional solidarity, quality of life, trust, and place attachment from the perspective of place identity and place dependence [14]. Therefore, it can be noted that previous studies have not integrated the sociopsychological constructs such as community attachment and involvement with residents' DSR as predictor factors of ERB. Thus, this study filled the void in the current literature by empirically validating the positive impacts of destination social responsibility on community attachment and involvement of the local residents of Ghana.

Finally, the study results contribute to the sustainable tourism development literature by assessing the mediation effect of community attachment and involvement through which DSR influences the ERB of local residents. The results of this study confirmed that community attachment, but not involvement, positively mediates the relationship between DSR and residents' ERB. Thus, previous studies validated the mediation effect of other environmental factors, such as tourism impact, resident identification, emotional solidarity, and place attachment [6,8,12,13,19,24,25]. For example, Hu et al. [12] argued that place attachment plays a mediating role through which DSR influences the ERB of local residents. Therefore, the results of this study closed the gap in the literature by responding to recent calls to investigate the intervening mechanism of community attachment and involvement through which environmental factors, such as DSR, influence different environmental outcomes, such as ERB, from the perception of local residents [6,14,146].

### 6.3. Practical Implications

The results of this study suggest several valuable practical and managerial implications for sustainable community-based tourism management. First, the results show that both community attachment and involvement are important outcomes of residents' DSR, which in turn leads to ERB. Thus, tourism destination management should promote the development of more DSR activities for residents to feel more attached to and involved in their local communities. Therefore, sustainable tourism managers should involve the local residents as primary stakeholders in developing and preparing tourism planning for their communities [187], and they should encourage them to engage in self-management activities, such as creating more opportunities for their local residents by increasing their participation in the decision-making processes [119,188] and promoting their tourism destinations in the social media marketing environment [189]. Moreover, increasing the involvement of local residents in different tourism management activities may allow them to be involved in all aspects of sustainable development toward their specific tourism destinations. Thus, improving residents' DSR practices would increase community involvement.

Second, the way local residents are attached to their communities depends greatly on the sense of belonging they feel toward their specific tourism destinations, and the role of community attachment is important in stimulating residents' DSR behaviors. Therefore, the results of this study revealed that residents' DSR increased community attachment, which in turn enhanced and improved their ERB activities. Accordingly, local authorities and destination managers may use this result to increase the sense of belonging and attachment between local residents and the community by increasing their DSR behaviors, which leads to enhancing and improving their ERB activities. Therefore, they can adopt programs such as satisfaction scores and feedback systems to give local residents a feeling of importance, increase their sense of belonging, and improve their ERB activities by experiencing a high level of attachment toward their local communities. When local residents are satisfied with their communities, they can also have a greater understanding of the social responsibility toward their tourism destination and feel a sense of the importance of a sustainable environment, which in turn promotes their ERB. Community attachment is an important factor in determining residents' ERB because it plays a substantial role in building and maintaining people-place relationships [12,75,127].

Finally, the results of this study showed that if destination managers wish to enhance and improve residents' ERB, they should adopt more DSR programs that promote the development of more sustainable destinations. Therefore, local authorities must encourage more DSR activities by allocating more resources and communicating their DSR activities with their residents using different forms of offline and online media, such as social media, which in turn leads to increases in their ERB. In addition, when local residents feel a sense of satisfaction and belonging to the environment, they become more attached to their communities. Local authorities should communicate environment-related problems and DSR activities to local residents, elaborating that their tourism destinations still need them to behave in a more environmentally responsible manner.

### 6.4. Limitations and Future Research Directions

In addition to the notable contributions outlined above, this study had several limitations. First, the lack of a significant impact of community involvement on residents' ERB was noteworthy. We call for future research to investigate the relationship between community involvement and residents' ERB using a long-term attitudinal research design. Second, to the best of the author's knowledge, no empirical study has been conducted to investigate the causal relationships between DSR, community attachment, community involvement, and resident ERB. The present study proposes a model that more accurately describes the effects of DSR on residents' ERB by introducing the possible mediation effect of community attachment and involvement in the model. Thus, future research should investigate the mediation effect of other factors (e.g., overall community satisfaction, sense of place, destination familiarity, and image) through which DSR influences residents' ERB in addition to community attachment and involvement [14].

Third, the use of the convenience sampling technique in this research allowed us to collect data from respondents by focusing on the three main tourist destinations in Ghana (Accra, Kumasi, and Sekondi-Takoradi), thus limiting the generalizability of the results to other tourist destinations in Ghana, such as Mole National Park and Tamale. In addition, this study was limited to one African country, Ghana; thus, the results cannot be generalized beyond the Ghanaian context. Therefore, further research is recommended to conduct surveys in other countries with cultures other than or similar to Ghana. Finally, a possible line of inquiry might investigate the impact of residents' DSR in the aftermath of health and social austerity induced by the COVID-19 outbreak [190,191], which "*have negatively affected all industries around the world including tourism*" [9]. This is an interesting way to validate the mediating role of community attachment and involvement in the relationship between DSR and residents' ERB during times of a particular type of environmental uncertainty.

**Author Contributions:** Conceptualization, E.N. and O.L.E.; methodology, E.N. and S.A.-G.; software, O.L.E.; validation, O.L.E. and H.Y.A.; formal analysis, H.Y.A.; investigation, E.N. and H.Y.A.; resources, E.N. and S.A.-G.; data curation, E.N.; writing—original draft preparation, E.N.; writing—review and editing, E.N., O.L.E. and H.Y.A.; visualization, O.L.E. and H.Y.A.; supervision, O.L.E.; project administration, O.L.E. and H.Y.A.; funding acquisition, E.N. and S.A.-G. All authors have read and agreed to the published version of the manuscript.

**Funding:** This research received no external funding.

**Institutional Review Board Statement:** Not applicable.

**Informed Consent Statement:** Informed consent was obtained from all subjects involved in the study.

**Data Availability Statement:** Not applicable.

**Acknowledgments:** The authors wish to thank the two anonymous reviewers for their constructive comments and the editor for his review of the manuscript. The authors would also like to thank all the local residents of Ghana for their time in participating in this research.

**Conflicts of Interest:** The authors declare no conflict of interest.

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
