# Peer review of "Destination Social Responsibility and Residents’ Environmentally Responsible Behavior: Assessing the Mediating Role of Community Attachment and Involvement"

_sustainability, doi:10.3390/su142114153_

Round 1
Reviewer 1 Report
Dear editor, thanks so much for giving me an opportunity to review this manuscript. Given that individual’s environmentally responsible behavior can significantly contribute to the Union 2030’s SDGs, the research on resident environmentally responsible behavior is a hot field for tourism academics. This paper aims to attempt the influence of DSR on RERB via the mediating role of community attachment and involvement. The review hypothesis is convinced, the data analysis method is scientific using SEM, and the conclusions and discussion are well written. Yet, there are some problems that should be paid attention to.
(1) Page 1, Line 1-2: Paper title should be revised. “residents environmentally responsible behavior” should be replaced by “resident environmentally responsible behavior” or “residents’ environmentally responsible behavior”.
(2) Page 4, Line 126- Page 5, Line 163: In this paper, environmentally responsible behavior is a very important term. As you know, the definition of environmentally responsible behavior is the equal to pro-environmental behavior. Several new important literature have been published in 2022 (Zheng et al., 2022). The mechanism of environmentally responsible behavior/Pro-environmental behavior is very important for this paper , so the authors should summary the related literature to present theory of planned behavior (Zheng et al., 2022), SOR theory, and NAM theory. You should present more systematically theory framework to review this issue. Only doing this, this paper’s academic value is a full explanation.
Here is a potential literature that can be included in the next revision as follows: (for instance)
Zheng, W.; Qiu, H.; Morrison, A.M.; Wei, W.; Zhang, X. Landscape and Unique Fascination: A Dual-Case Study on the Antecedents of Tourist Pro-environmental Behavioral Intentions. Land 2022, 11, 479.
(3) Page 4, Line 164-Line 176: In this paper, SOR theory is a very framework for exploring RERB. However, why choose SOR as the basic framework? Why choose attachment and involvement as organism? It is a pity that I can not find the reasonable explanation for the above re-solved problem.
Here is an important literature (Qiu et al., 2022) that employs SOR theory to explore the internal mechanism of environmentally responsible behavior via the mediating role place attachment. This literature can provide tenable evidence for the mediating role of place attachment within SOR framework.
Qiu, H.; Wang, X.; Wu, M.-Y.; Wei, W.; Morrison, A.M.; Kelly, C. The Effect of Destination Source Credibility on Tourist Environmentally Responsible Behavior: An Application of Stimulus-Organism-Response Theory. J. Sustain. Tour. 2022, 1–21. https://doi.org/10.1080/09669582.2022.2067167
(4) where are scale indicators? Please provide detail indicators of this scale in the revision.
(5) the normality test is omitted. Please add it.
(6) common method variance (CMV) test is also omitted. Please include in the next revision version.
(7) the reference format is needed to be adjusted to meet SUS’s requiments. Please check it carefully!
In fact, all of the above comments contribute to enhance the paper quality. I wish to review the authors’ next revised manuscript in the future.
Good luck!
Author Response
We appreciate the editor and reviewers’ time to read the script and make these comments. We have taken every step to satisfy the editor and reviewers’ comments and feedback as shown below. The revised script has all changes highlighted in blue for easier tracking of changes made. Please kindly check the detailed doc file. Everything is self-explanatory in that file.

Reviewer 2 Report
Thanks for giving the opportunity to review the paper titled as Destination social responsibility and residents environmentally responsible behavior: Assessing the mediating role of community attachment and involvement. This paper is timely, well developed, conducted and well written. Thus, it addresses a significant topic likely to be of interest to tourism destination sustainalbe development. The concept of destinaiton social responsibility (DSR) was formally proposed by Su and colleagues., and it has been attracted extensive attention by tourism scholars in recent years. Compared to CSR, DSR considers the complex nature of destination, and diverse stakeholders, which leads to more suitable for tourism destination level.
This paper examined the relaitonship among DSR, community attachment, involvement, and residents environmentally responsible behavior, which has valued for destination sustainable development and resident quality of life. Thus, the reviewer consider it is a valuable, original article, and recommend to accept it for publication.
Author Response

(The authors gave the same response as above.)

Reviewer 3 Report
The article was prepared with great concern. The authors have analyzed the available literature on the subject, set the research questions well and made hypotheses. Verification of hypotheses on a large group of respondents, however, the return at the level of 49% may represent a large statistical error and lack of representativeness. The text does not specify the size of the research samples.
Well selected statistical tools for calculating data.
Well-directed discussion and conclusions drawn.
Author Response

(The authors gave the same response as above.)

Round 2
Reviewer 1 Report
Congratulation! You got it!